# Personality and emotional intelligence of researchers: The importance of affects

**Laura Hernando-Jorge**[1]*, **Anabel Fernández-Mesa**[2], **Joaquín M. Azagra-Caro**[3], **Ana M. Tur-Porcar**[1]

1 Faculty of Psychology, Universitat de València, Valencia, Spain, 2 Faculty of Economics, Universitat de València, Valencia, Spain, 3 INGENIO (CSIC-Universitat Politècnica de València), Valencia, Spain

* lauher9@alumni.uv.es

**Data Availability Statement:** Data cannot be shared publicly because of the confidentiality agreement signed by the participants. Data are available from the INGENIO (CSIC-UPV) Institutional Data Access (contact via

## Abstract

Researchers, who play a crucial role in knowledge production, deal with various emotions in their challenging work environment. Their personality might affect how well they manage their emotions, but their moods could help counteract these effects. This study aims to investigate whether researchers' moods influence the connection between their personality and emotional intelligence. 7,463 Spanish researchers replied to an online survey. Responses analysed through partial least squares structural equation modelling show significant positive relationships between the big five personality traits (openness to experience, conscientiousness, extraversion, agreeableness and emotional stability) and emotional intelligence. In addition, positive affect positively mediates the relationships between each of the personality traits and emotional intelligence, and negative affect mediates the same relationships but negatively. The importance of managing emotional states to regulate emotional experiences in the work of researchers is discussed.

## Introduction

Emotions are integral to the functioning of organizations, influencing factors such as stress resilience, commitment, and performance [1–3]. Within the research community, individuals contend with a multitude of emotions in a demanding professional environment. This atmosphere, characterized by a focus on scientific impact, often leads to work overload, intense competition, reputation concerns, and a reluctance to acknowledge challenges or dissatisfaction, which can adversely affect well-being and mental health [4].

At the core of managing these emotional complexities lies emotional intelligence (EI), essential for regulating one´s emotions and navigating challenging situations [1]. In particular, researchers must constantly manage intense feelings of excitement, frustration, and self-doubt as they pursue discoveries, secure funding, and face the peer review process. Possessing high EI can help researchers maintain focus, foster productive collaborations, and persevere through the setbacks inherent in scientific research [5].

Despite the importance of EI in research settings, our understanding of researchers' emotional intelligence remains limited [3], failing to reflect the societal significance of these

biblioteca@ingenio.upv.es) for researchers who meet the criteria for access to confidential data.

**Funding:** The Ministry of Science and Innovation funded this research through Project CSO2016-79045-C2-2-R of the Spanish National R&D&I Plan. Funding is also available from the Conselleria de Innovación, Universidades, Ciencia y Sociedad Digital, Generalitat Valenciana, AICO/2021/021; and from Ministerio de Ciencia, Innovación y Universidades, PID2022-137053NB-100. The funders had no role in the study design, data collection and analysis, decision to publish, or preparation of the manuscript.

**Competing interests:** The authors have declared that no competing interests exist.

knowledge generators. Researchers often exhibit unique personality traits, such as introversion and neuroticism [6, 7], which may pose challenges in developing and utilizing emotional intelligence. Addressing this gap in the literature is essential, as enhancing researchers' EI could lead to improved well-being, increased research productivity, and more impactful scientific contributions.

The primary research question of this study is therefore: Does researchers' personality influence their EI? Indeed, personality traits are known to underpin emotional experiences [8]. Individuals scoring high in traits such as extraversion or emotional stability may naturally exhibit heightened emotional intelligence [9, 10]. However, previous research has yet to explore this relationship within the context of researchers.

The secondary research question is: Can researchers' affect mitigate the potential impact of their personality on EI? Positive and negative affect represent transient emotional states and are therefore more amenable to change than stable personality traits [11]. If affect serves as a mediator between personality and EI, targeted interventions could potentially offset the constraints imposed by researchers' personality traits and bolster their EI. Furthermore, this question extends beyond the case of researchers, as the broader literature on EI, to which this study contributes, has yet to investigate this model linking personality, affect, and EI.

With these factors in consideration, the objective of this study is to analyse whether researchers' affect moderates the relationship between personality and EI. By achieving this objective, our understanding of researchers' emotional experiences will take a modest stride forward, potentially empowering better-regulated researchers to enhance their scientific performance and impact.

## Literature review

### Personality traits and emotional intelligence

Personality traits play a fundamental role in shaping the fundamental and enduring characteristics of individuals, influencing their thoughts, feelings, and behaviours over time [12, 13]. These traits correspond to people's basic characteristics, relatively stable, which, combined with adaptations to the environment, gradually create behavioural patterns [12].

The Five-Factor Model of Personality [12, 14] is widely recognized among the scientific community in light of the preceding research. It includes the Big Five personality traits, identified as the major dimensions or traits that capture the most important variations in personality [14]: Openness to experience involves being imaginative, creative, and curious, fostering the ability to contribute original ideas while embracing diverse experiences. Conscientiousness involves organization, responsibility, and achievement orientation, reflecting thoroughness and goal-directed behaviour. Extraversion denotes a preference for social relationships, showing sociability, assertiveness and enthusiasm in individuals' behaviour. Agreeableness also reflects a tendency towards friendly social relationships, showing kindness, empathy and concern for the welfare of others. Finally, emotional stability (or its opposite, neuroticism) is characterised by calmness and balance, showing resilience and emotional equilibrium in various situations [14, 15]. Conservation of resources theory (COR) proposes that personality can act as a personal resource, enabling employees to respond effectively to situational demands and acquire additional resources [16, 17]. Thus, all these personality traits within the scientific community contribute significantly to achieving excellence and high performance in their work [18, 19]. In particular, high scores in openness to experience, conscientiousness, extraversion and emotional stability have shown positive effects on academic performance in higher education [15, 20, 21] as well as better work performance in different fields [19, 22]. Lounsbury et al.'s [6] comparison of personality traits between scientific and non-scientific population

revealed that scientists obtain higher scores in openness but lower ones in conscientiousness, extraversion and emotional stability. Indeed, researchers generally show a greater disposition towards opening up to new experiences and research [6], with a tendency towards introversion, although those who are more extroverted tend to express greater job satisfaction [23]. Personality traits shed light on researchers' perceptions of the impact of their work on academic, business or social beneficiaries, highlighting the potential for conflict when attempting to simultaneously benefit multiple groups at the same time [24].

Personality factors extend beyond individual performance, serving also as indicators of social effectiveness and of fostering social networks [25]. Forming networks is crucial in the scientific community to improve individual performance and strengthens collaborations [21, 26], although care must be taken to avoid the risk of attracting collaborators in excessive numbers or of lower quality [27]. Given the importance of sociability in the dynamics of research, it seems necessary to take into account not only personality traits, but also the emotional processes intrinsically linked to social functioning in the pursuit of scientific excellence [28, 29].

Emotional intelligence (EI) emerges as an important facet, understood as both a skill and a trait. As a skill, EI refers to the cognitive abilities required to understand one's own and others' emotions, regulate them, use emotional information to guide thoughts and actions, and thus to manage those emotions optimally [30, 31]. As a trait, EI encompasses the emotional self-perceptions in the lower levels of personality [32]. These two theoretical perspectives converge, highlighting the fundamental role of EI in the understanding and managing emotional perceptions, which are essential emotional skills for adapting to the demands of daily life. Therefore, it is convenient that they are present in the scientific population [2, 3, 33].

Empirical evidence indicates high correlations between some personality traits (such as conscientiousness and extraversion) and EI, especially when mixed measurements of EI are used [9, 10] that are closer to emotional competence [34]. Moreover, it has been verified that personality traits are important predictors of EI, since they explain 50% of the variance [35, 36]. So, based on these findings and the previous research, this is the first hypothesis:

H1. Researchers' personality traits (openness to experience, conscientiousness, extraversion, agreeableness and emotional stability) will be positively related to EI.

The demanding nature of scientific work, characterized by continuous research and publication is fraught with uncertainty and frustrations. This underlines the importance of taking into account factors that facilitate emotional understanding in the pursuit of optimal mood management [37, 38]. The next section deepens into the role of positive and negative affective emotions and their connection to personality traits and emotional intelligence.

## Positive and negative affective emotions, personality traits and emotional intelligence

Affective emotions encompass an individual's positive or negative mood, made up of both positive emotions or positive affect (ex: joy) and negative emotions or negative affect (ex: anger) [39]. Positive and negative emotions are complex, individual and multi-systemic responses that occur during the assessment or interpretation of events [11], with personality traits influencing these affective reactions [40].

According to Fredrickson's broaden-and-build theory [11], positive emotions play a crucial role in building resources to regulate negative emotional experiences in daily life, fostering recovery and counteracting the physiological effects associated with negative affect. Indeed, positive affect improves behavioural flexibility, increases attention and fosters well-being, which contributes to improve job performance, in addition to an increase in positive

interpersonal and task-related work events [39, 41–44]. Conversely, negative emotions are autonomously activated and reduce people's behavioural repertoire, negatively impacting productivity and predicting, in turn, an increase in negative interpersonal and task-related work events [11, 43].

The relationship of personality traits to affective well-being is well documented. A large body of research establishes a positive correlation between personality traits and well-being. Personality is a consistent predictor of affective well-being (affect) and not so much of cognitive well-being (life satisfaction) [35, 40, 42, 45–47]. Specifically, it was found that neuroticism (and its opposite, emotional stability) and extraversion were the strongest predictors of negative and positive affect, respectively. The other personality traits were also associated with both types of affect, although in a weaker way [48]. Based on these results, the following hypotheses are formulated:

H2a. Researchers' personality traits will be positively related to positive affect.

H2b. Researchers' personality traits will be negatively related to negative affect.

Previous literature also has verified connections between EI and positive or negative affect. EI helps to cope with emotions harmoniously because it fosters a more adaptive kind of emotion regulation and helps to adopt coping strategies that lead to events being experienced in a more positive way [38, 40, 46, 47, 49]. Thus, people with higher EI can manage emotions effectively, understand others' emotions accurately and experience more positive affect and less negative affect [46, 49, 50]. Positive affect optimises emotional performance by broadening the repertoire of thoughts and actions, in contrast to negative affect, which in turn can benefit work performance [35, 44]. The following hypotheses are proposed:

H3a. Researchers' positive affect will be positively related to EI.

H3b. Researchers' negative affect will be negatively related to EI.

In general terms, personality traits play a determining role in emotional competence [10, 34] as well as emotional-affective reactions to events, encompassing both positive and negative aspects [40]. To a large extent, personality traits are predictors of EI, and the ability to experience positive and negative emotions is known to provide an occupational advantage, as it helps to develop personal, psychological, intellectual and social resources [1, 11, 51]. Specifically, positive emotions tend to improve behavioural flexibility and to counteract negative emotions, thereby increasing the behavioural repertoire [43]. Therefore, affect may provide a mechanism by which personality traits influence emotional intelligence. Consequently, experiencing a higher level of positive affect will be a mechanism that allows researchers to improve their emotional intelligence, whereas experiencing higher level of negative affect will decrease their emotional intelligence. Therefore, the hypotheses would be:

H4a. Researchers' positive affect positively mediates the relationship between personality traits and emotional intelligence.

H4b. Researchers' negative affect negatively mediates the relationship between personality traits and emotional intelligence.

To sum up, this study explores the interaction between individual and social factors, specifically with the aim of analysing the relationships among personality traits, positive and negative affect and EI among scientists engaged in advanced knowledge. Based on this review of the literature, Fig 1 shows a diagram of the proposed theoretical model to be empirically tested.

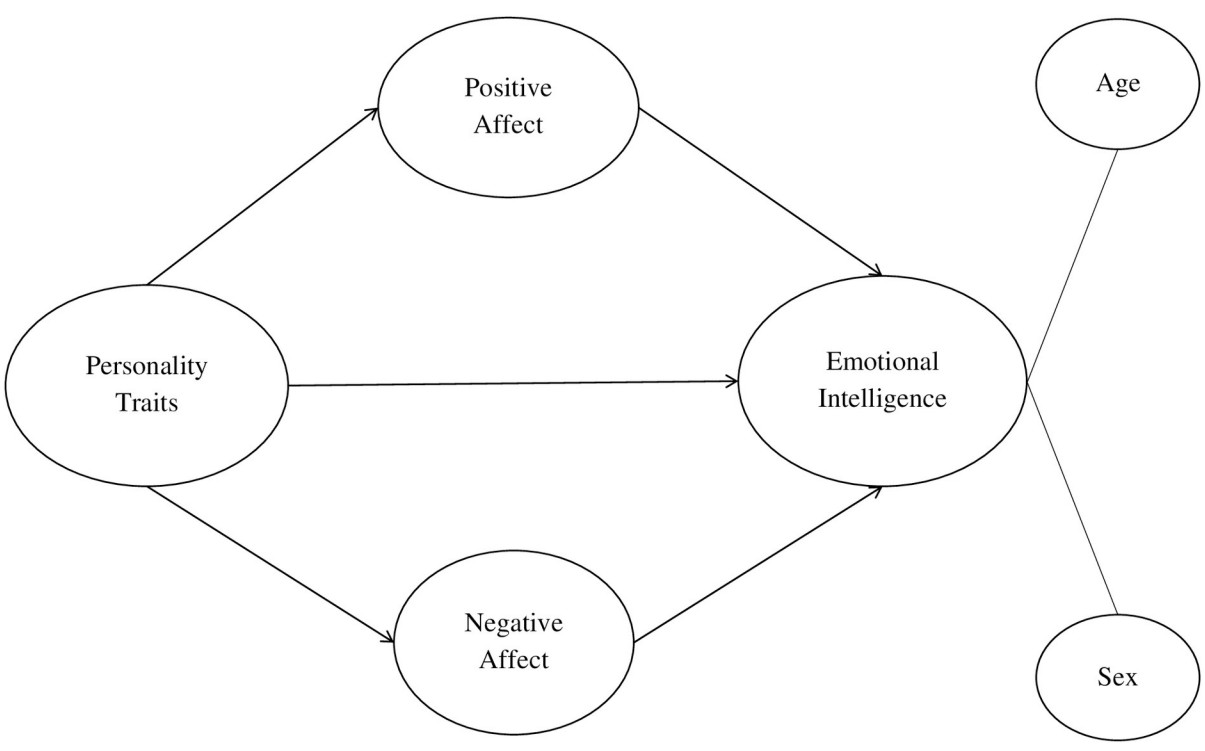

**Fig 1. Theorized mediation model.**

## Materials and methods

The study objective requires finding empirical support to the theoretical relationships established among clearly predefined variables, based on prior literature. To that end, a standard approach is followed. First, a quantitative design is developed to quantify relationships between variables, providing numerical evidence of the strength and direction of associations. A qualitative design would be more appropriate if the objective of the research was to create new theory, but when the objective is to generalize a phenomenon previously verified through case studies or induced from previous literature, quantitative research is more appropriate. Second, a survey was conducted, because there is no public information available on the researchers' characteristics needed for this study. Third, structural equation modelling (SEM) was performed, to test the model and provide scientific generalizable evidence of how to increase (decrease) emotional intelligence through positive (negative) affects and personality traits. SEM allows for the analysis of complex relationships between multiple variables, related to each other through regression coefficients. SEM is particularly suited to implement mediation analysis through a system of equations.

### Design of the study

The study population is Spanish researchers, defined as authors of scientific publications, affiliated to a Spanish organization and taken from the corresponding authors on publications in the Web of Science (WoS) listings from 2013 to 2016. The corresponding author is usually one of the lead authors and lead contributors to the content of the manuscript [52] or is perceived as so [53]. Editors appoint corresponding authors as reviewers [54] and they are considered reliable sources of knowledge about the publication and the underlying research [55].

We gathered about 65,000 valid e-mails. Helsinki ethical standards for research involving human beings were followed, e.g., respondents were informed of the objectives of the study, the anonymization process, the safeguards to encrypt their data and the possibility to withdraw their responses, before asking for their written informed consent. The project received a favourable report from the Ethics Committees of the Spanish National Research Council (CSIC) and of the Polytechnic University of Valencia. Before sending the survey to the entire identified population, a pilot study was performed. To do this, in July 2017 we sent the survey to 62 researchers and in April 2018 we sent a second one to 56 researchers, in order to analyse the responses and adjust the final survey. In November 2018, the final survey was sent to the 65,000 identified researchers, via a link using an online survey management software.

Three reminders were sent to get more participants. Finally, 7,463 gave valid answers, that is, a response rate of 11.48% and a sample size with a 95% confidence level and a 1% margin of error, which is representative of the population. Those are the responses that have been included in the statistical analyses.

The final distribution was as follows: 57% men and 43% women. The participants were aged between 20 and 96 years (M = 48.76, SD = 10.238). 73% worked for universities or research organisations, of which 46% were tenured researchers. The rest worked for health establishments (16%), business firms (4%), public administrations (4%) and non-governmental organisations (3%). See full demographics in [56]. The distribution by science field is as follows: technological sciences (14%), medical sciences (24%), life sciences (16%), natural sciences (24.48%) and social sciences (21%). This broad representation of researchers underscores the diversity of participants in the study.

The researchers taken into account in the study filled in all of the questionnaires about psychological variables, though a small percentage did not answer the questions about sex or age, who have been labelled as "Not known".

## Measurements

**Emotional intelligence.**   A Spanish adaptation by Carvalho et al. [37] of Wong Law Emotional Intelligence Scale (WLEIS) [57] was used. It evaluates EI through statements that the person must assess and then decide if they correspond to their situation. A Likert scale was used with seven possible answers (1 = completely disagree; 7 = completely agree). Factors: evaluation of one's own emotions, evaluation of the emotions of others, use of emotion and emotion regulation. In this study, EI has been considered as a single factor, following the concept of Wong and Law [57] when they affirm that people with higher levels of EI in general can make use of their emotion regulation mechanisms effectively to create positive emotions and foster emotional and intellectual growth. Example of an item: "I can control my temperament to tackle difficulties rationally." Cronbach's alpha was .90.

**Personality.**   A Ten-Item Personality Inventory or TIPI [58] was used. This is a Spanish adaptation of Renau et al. [59] and of Romero et al. [60]. Both overlapping translations are used in order to improve the psychometric properties of the scale. It evaluates personality traits according to the Big Five model [61] using 14 items. The factors are: openness to experience (e.g., "curious, multi-faceted"), conscientiousness (e.g., "reliable, self-disciplined"), extraversion (e.g., "extroverted, enthusiastic"), agreeableness (e.g., "considerate, affectionate") and emotional stability (e.g., "calm, emotionally stable"). A Likert scale was used with seven possible answers (1 = completely disagree; 7 = completely agree). Cronbach's alphas were: openness to experience ($\alpha$ = .74), conscientiousness ($\alpha$ = .62), extraversion ($\alpha$ = .71), agreeableness ($\alpha$ = .68) and emotional stability ($\alpha$ = .67). Alphas greater than .60 can be considered adequate [60, 62, 63].

**Positive and negative affect.** The International Positive and Negative Affect Schedule Short Form (I-PANAS-SF) [64] was used in a Spanish adaptation by López-Gómez et al. [65] corresponding to the 10 items of the I-PANAS-SF. It evaluates the level of positive affect (e.g., "inspired") and negative affect (e.g., "aggressive") via different moods. A Likert scale of five alternatives was used (1 = Never and 5 = Always). Cronbach's alphas were: positive affect, α = .76; negative affect, α = .67.

**Control variables.** Previous research has shown that sex and age have a significant effect on emotional competence. It is consistent in the literature that women have higher EI [32, 35, 37, 66] and that this variable may improve with age [66, 67]. In addition, women also experience higher levels of both positive and negative affect [41, 68, 69] and some studies have also found significant relationships between age and negative affect [64, 65]. Therefore, they were included as control variables, to control for possible effects of sex or age differences. Sex was coded as 0 = male, 1 = female. Age was calculated in logarithm.

## Data analyses

The data analyses in this study primarily rely on the application of SEM, which has been widely used across various academic disciplines to substantiate theory. SEM involves constructing measurement models to define latent variables and establishing relationships through structural equations. To estimate the models for research hypotheses, PLS-4 software was utilized. Additionally, confirmatory factor analysis (CFA) was employed to assess the quality of the measurement scales. A commonly recommended guideline for implementing SEM is to have a minimum sample size of 100 subjects [70], which our sample surpasses.

## Results

### Psychometric properties of the measurement scales

Table 1 presents the correlations between the variables included in the empirical analysis, along with the corresponding descriptive statistics.

In this study reflective measures are used and there are four criteria to assess a reflective measurement model in PLS: factor loading, composite reliability (CR), average variance extracted (AVE) and discriminant validity [71].

**Table 1. Means, standard deviations and correlations among study variables.**

|  | 1 | 2 | 3 | 4 | 5 | 6 | 7 | 8 | 9 |
|---|---|---|---|---|---|---|---|---|---|
| 1. Openness to experience | - | | | | | | | | |
| 2. Conscientiousness | .148** | - | | | | | | | |
| 3. Extraversion | .328** | .128** | - | | | | | | |
| 4. Agreeableness | .244** | .253** | .102** | - | | | | | |
| 5. Emotional Stability | .149** | .218** | .071** | .495** | - | | | | |
| 6. Emotional Intelligence | .392** | .373** | .318** | .415** | .420** | - | | | |
| 7. Positive Affect | .437** | .366** | .341** | .210** | .241** | .521** | - | | |
| 8. Negative Affect | -.134** | -.177** | -.135** | -.252** | -.525** | -.290** | -.221** | - | |
| 9. Age | .020 | -.045** | .007 | -.056** | .042** | -.014 | .034** | -.152** | - |
| 10. Sex | -.024* | .198** | .141** | .083** | -.044** | .119** | .103** | .079** | -.152** |
| Mean | 5.26 | 5.25 | 4.59 | 5.30 | 4.97 | 5.06 | 3.94 | 2.41 | 48.76 |
| Standard Deviation | .871 | 1.070 | 1.248 | .845 | 1.100 | .677 | .467 | .502 | 10.238 |

Note: To calculate the correlation coefficients, authors worked with the means of the items that make up each dimension.

*p < .01

** p < .001

**Table 2. Measurement model results.**

|  | Factor loading | Cronbach's α | CR | AVE |
|---|---|---|---|---|
| **Openness to experience** | [.702 - .798] | .735 | .834 | .558 |
| **Conscientiousness** | [.818 - .882] | .622 | .840 | .724 |
| **Extraversion** | [.804 - .936] | .705 | .864 | .762 |
| **Agreeableness** | [.604 - .802] | .683 | .801 | .507 |
| **Emotional Stability** | [.854 - .881] | .673 | .859 | .753 |
| **Positive Affect** | [.615 - .792] | .762 | [.840 - .841] | [.513 - .514] |
| **Negative Affect** | [.436 - .793] | .671 | [.777 - .791] | [.421 - .435] |
| **Emotional Intelligence** | [.500 - .767] | .901 | [.915 - .916] | [.405 - .407] |

Note: statistics for some variables ranged across the five models, so they are indicated as a range in the table.

Factor loading. Carmines and Zeller [72] recommend factor loadings of the constructs be equal to or greater than .7, since this is the level at which 50% of the variance of the indicator is explained by its factor. If the loading is between .40 and .70 it should be checked whether the elimination of this indicator improves the composite reliability [62]. The factor loadings of all the variables can be consulted in Table 2. In the case of the openness to experience, responsibility, extraversion and emotional stability items, their factor loadings were all above the recommended level. However, there were two items of agreeableness and positive affect with loadings below the minimum level, but which did not improve the reliability of the models after their elimination. In addition, several items of negative affect and EI also showed loadings below the minimum level, but again their elimination did not improve the reliability values.

Composite reliability. The use of the Cronbach's alpha coefficient in isolation is not recommended to evaluate the reliability of a measurement scale [73] and that it is why in this study it is also presented composite reliability. This measure considers that indicators present different loadings and their value should be higher than .6 [60, 62]. The composite reliability of the constructs is shown in Table 2, exceeding in all cases the recommended minimum.

Convergent validity. This criterion ensures that a set of indicators represent the same underlying factor [74]. The AVE value must be at least .500 [75], which implies that the factor explains, on average, more than half of the variance of its items. The AVE values can be found in Table 2. The elimination of items with lower external loadings did not increase this index in the two constructs that did not exceed the recommended minimum (negative affect and EI), so no changes were made.

Discriminant validity. In order to test discriminant validity, to evaluate whether the indicators are unrelated to those factors that are known to be independent of the variable they are intended to measure, the criteria of Fornell and Larcker were used [74]. These authors recommend that the square root of the average variance extracted (AVE) should be greater than the correlations between a construct and the other constructs. In all five models, indeed, the criterion is met (Table 3).

Henseler and company [76] developed a new method for assessing discriminant validity, heterotrait-monotrait (HTMT). This criterion would indicate that there is discriminant validity when correlations between constructs are less than .70. The correlations ranged, in all models, between .314 (negative–positive affect relationship) and .629 (EI–positive affect relationship), except in the emotional stability–negative affect relationship (.776), so the criterion is partially met (Table 4).

**Table 3. Discriminant validity analysis—Fornell-Larcker criterion.**

| Openness to experience | 1 | 2 | 3 | 4 |
|---|---|---|---|---|
| 1. Negative affect | (.652) | | | |
| 2. Positive affect | -.241 | (.716) | | |
| 3. Openness to experience | -.165 | .458 | (.747) | |
| 4. Emotional intelligence | -.290 | .539 | .429 | (.637) |
| **Conscientiousness** | 1 | 2 | 3 | 4 |
| 1. Negative affect | (.657) | | | |
| 2. Positive affect | -.241 | (.717) | | |
| 3. Conscientiousness | -.162 | .387 | (.851) | |
| 4. Emotional intelligence | -.296 | .537 | .406 | (.636) |
| **Extraversion** | 1 | 2 | 3 | 4 |
| 1. Negative affect | (.649) | | | |
| 2. Positive affect | -.249 | (.716) | | |
| 3. Extraversion | -.180 | .381 | (.873) | |
| 4. Emotional intelligence | -.292 | .539 | .362 | (.637) |
| **Agreeableness** | 1 | 2 | 3 | 4 |
| 1. Negative affect | (.657) | | | |
| 2. Positive affect | -.224 | (.717) | | |
| 3. Agreeableness | -.239 | .261 | (.712) | |
| 4. Emotional intelligence | -.296 | .526 | .497 | (.638) |
| **Emotional stability** | 1 | 2 | 3 | 4 |
| 1. Negative affect | (.660) | | | |
| 2. Positive affect | -.210 | (.717) | | |
| 3. Emotional stability | -.534 | .248 | (.868) | |
| 4. Emotional intelligence | -.307 | .526 | .460 | (.636) |

Note: Diagonal elements (in parenthesis) are the square root of the AVE; off-diagonal elements are the correlations among constructs in the inner model.

## Evaluation of the structural model

Bootstrapping procedure was used for the estimation of the model parameters by minimizing their standard errors [77]. Following the recommendations of Chin [78], the bootstrap estimates presented in this article are based on 5000 bootstrap samples. The fundamental criteria for evaluating the structural model have been the $R^2$ values (coefficient of determination) of the endogenous latent variables and the strength of the relationship between factors [75].

Figs 2–6 show the values and significance of the path coefficients and the $R^2$ of the endogenous variables of the model for openness to experience, responsibility, extraversion, agreeableness and emotional stability, respectively.

In relation to the models themselves, as can be seen in Table 5 and as could be extracted from the previous results also, after controlling for the variables age and sex, a significant positive effect of the five personality traits, namely openness to experience, conscientiousness, extraversion, agreeableness and emotional stability, of the researchers on their emotional intelligence was found; H1 is therefore supported. On the other hand, it has also been found that positive affect is positively and significantly related to personality traits in the 5 models, and negative affect is negatively and significantly related to them, confirming H2a and H2b. Finally, in terms of hypotheses, it can be seen that positive affect was positively related to EI in all models, and negative affect was inversely related to EI in all of them, also confirming H3a and H3b.

**Table 4. Discriminant validity analysis–HTMT.**

| Openness to experience | 1 | 2 | 3 |
|---|---|---|---|
| 1. Negative affect | | | |
| 2. Positive affect | .314 | | |
| 3. Openness to experience | .202 | .598 | |
| 4. Emotional intelligence | .377 | .629 | .509 |
| Conscientiousness | 1 | 2 | 3 |
| 1. Negative affect | | | |
| 2. Positive affect | .314 | | |
| 3. Conscientiousness | .261 | .545 | |
| 4. Emotional intelligence | .377 | .629 | .518 |
| Extraversion | 1 | 2 | 3 |
| 1. Negative affect | | | |
| 2. Positive affect | .314 | | |
| 3. Extraversion | .233 | .475 | |
| 4. Emotional intelligence | .377 | .629 | .414 |
| Agreeableness | 1 | 2 | 3 |
| 1. Negative affect | | | |
| 2. Positive affect | .314 | | |
| 3. Agreeableness | .368 | .329 | |
| 4. Emotional intelligence | .377 | .629 | .589 |
| Emotional Stability | 1 | 2 | 3 |
| 1. Negative affect | | | |
| 2. Positive affect | .314 | | |
| 3. Emotional stability | .776 | .339 | |
| 4. Emotional intelligence | .377 | .629 | .545 |

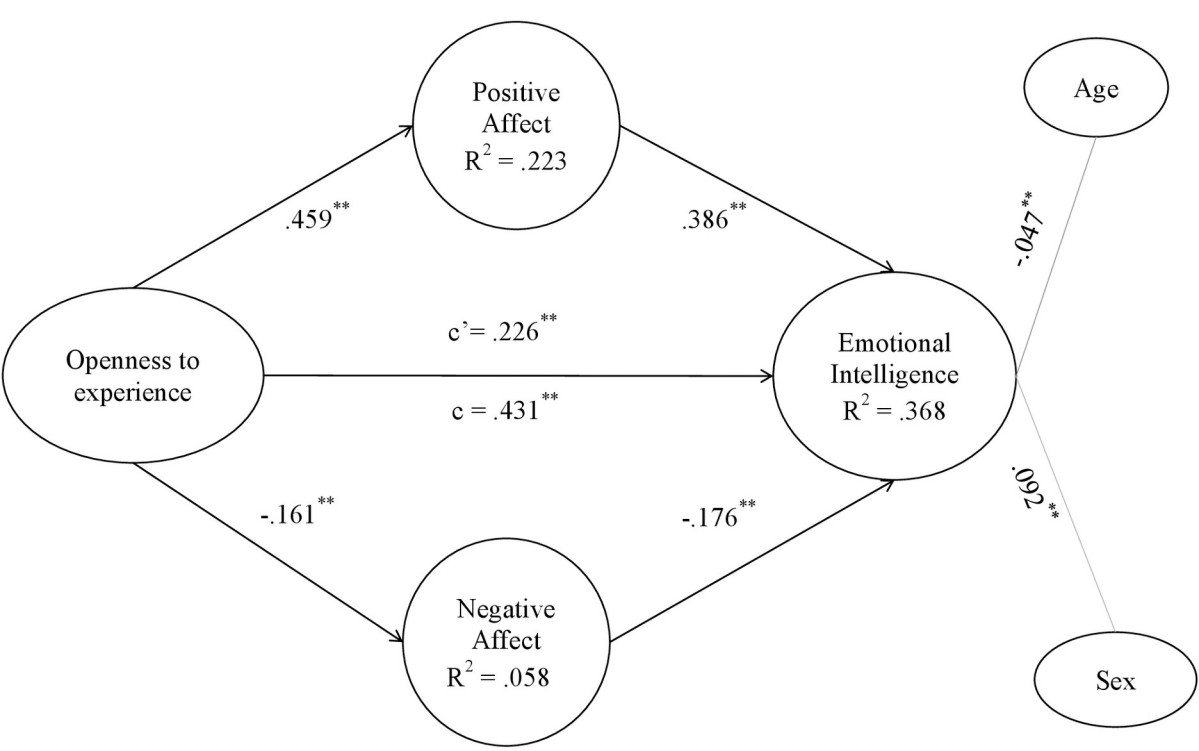

**Fig 2. Openness to experience model.** Note: c' = direct effect; c = total effect. $^{**}p < .001$; $^{*}p < .01$.

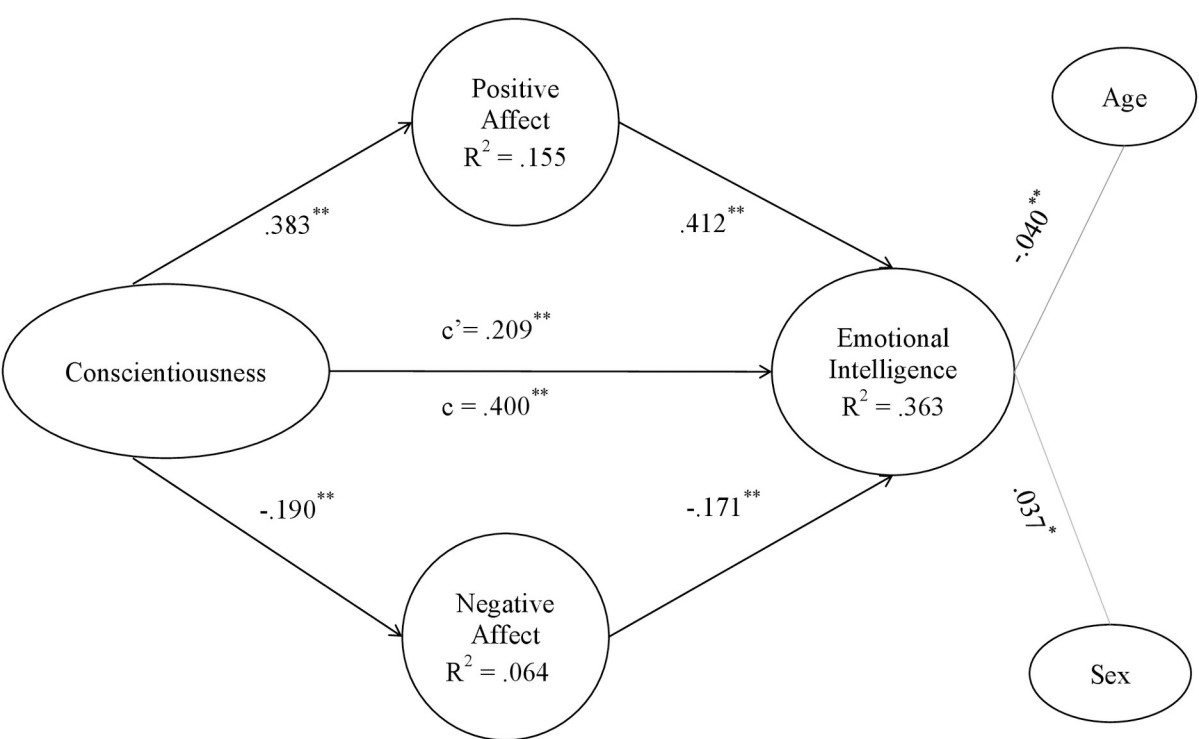

**Fig 3. Conscientiousness model.** Note: c' = direct effect; c = total effect. **p < .001; *p < .01.

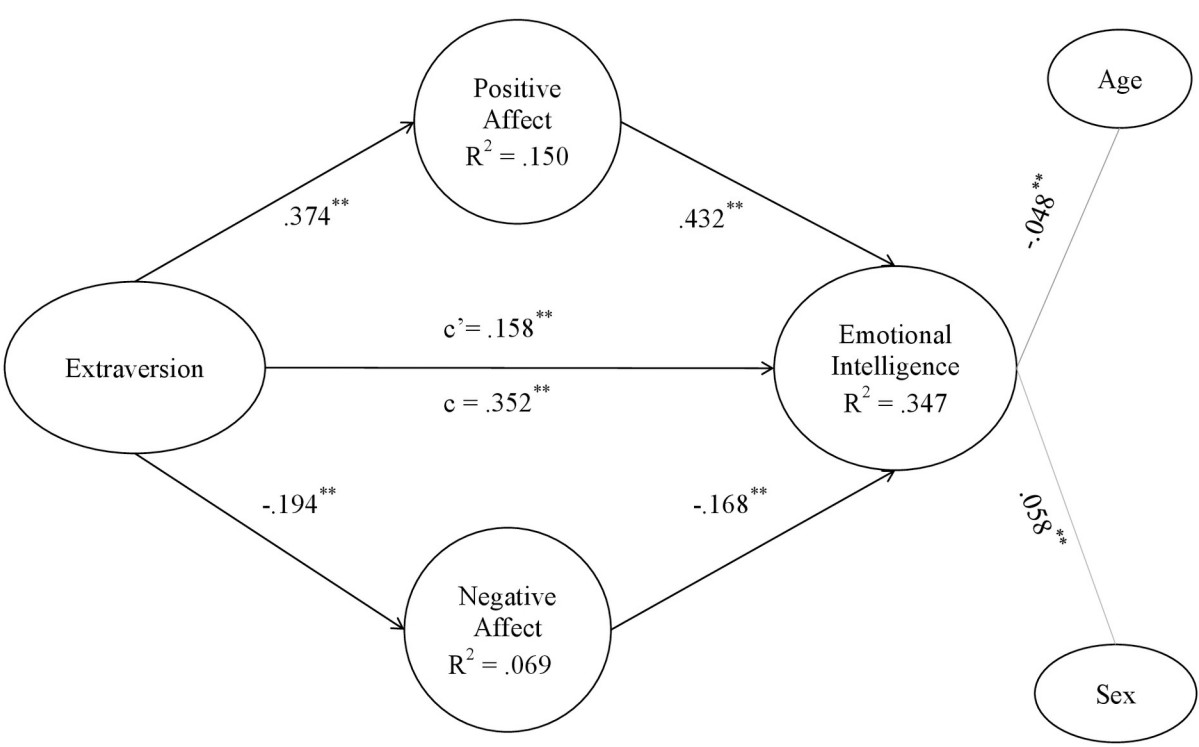

**Fig 4. Extraversion model.** Note: c' = direct effect; c = total effect. **p < .001; *p < .01.

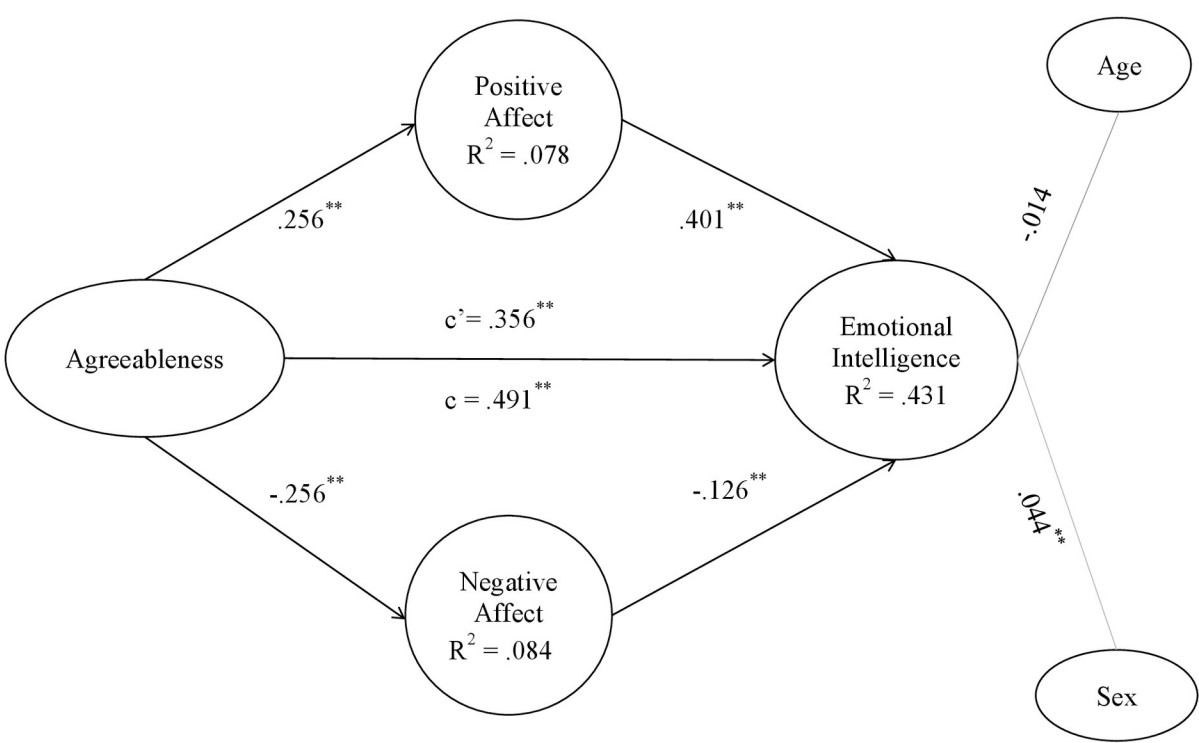

**Fig 5. Agreeableness model.** Note: c' = direct effect; c = total effect. **p < .001; *p < .01.

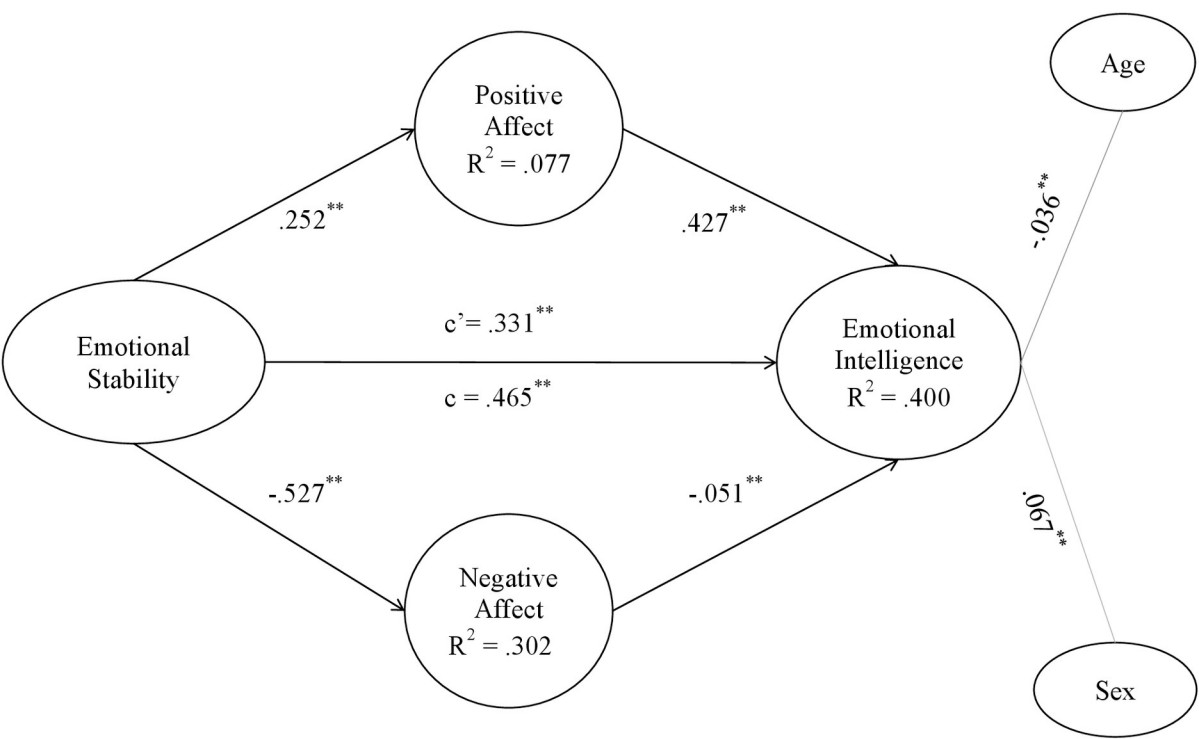

**Fig 6. Emotional stability model.** Note: c' = direct effect; c = total effect. **p < .001; *p < .01.

To test the mediating role of affect in personality—EI relationship, as established in the proposed theoretical model, bias-corrected bootstrapping was employed to generate 95% confidence intervals around the indirect effect of both positive and negative affect [79, 80]. The analysis, performed with 5000 bootstrap samples, revealed significant indirect effects of positive and negative affect on the personality traits—EI relationship, as the confidence intervals (95% CI) did not include 0 (see Table 5).

As shown in Figs 2–6 and Table 6, the $R^2$ index shows that the theoretical models explain at least 35% of the variability of EI in the five different models. This index should be greater than

**Table 5. Analysis of the mediating effect.**

| | | Direct models | | Mediated models | | | | | | | |
|---|---|---|---|---|---|---|---|---|---|---|---|
| | | Total effect (c) | t-value | Total effect (c) | t-value | Direct effect (c') | t-value | Indirect effect (a.b) | | t-value | LLCI | ULCI |
| Openness to experience model | O → EI | .447** | 48.09 | .431** | 42.25 | .226** | 19.84 | PA: | .177** | 28.58 | .165 | .190 |
| | | | | | | | | NA: | .028** | 10.02 | .023 | .034 |
| | | | | | | | | Total: | .206** | 30.76 | .193 | .219 |
| | Sex → EI | .151** | 13.92 | .126** | 11.30 | .092** | 9.24 | - | | - | - | - |
| | Age→ EI | -.003 | .28 | -.003 | .26 | -.047** | 4.91 | - | | - | - | - |
| | PA → EI | - | - | .386** | 35.39 | .386** | 35.39 | - | | - | - | - |
| | NA → EI | - | - | -.176** | 14.51 | -.176** | 14.51 | - | | - | - | - |
| | O → PA | - | - | .459** | 45.89 | .459** | 45.89 | - | | - | - | - |
| | O → NA | - | - | -.161** | 12.72 | -.161** | 12.72 | - | | - | - | - |
| Conscientiousness model | C → EI | .409** | 39.27 | .400** | 36.43 | .209** | 18.24 | PA: | .158** | 25.99 | .146 | .170 |
| | | | | | | | | NA: | .033** | 10.30 | .027 | .039 |
| | | | | | | | | Total: | .190** | 27.72 | .177 | .204 |
| | Sex → EI | .053** | 4.37 | .037* | 3.19 | .037** | 3.64 | - | | - | - | - |
| | Age→ EI | .011 | 1.05 | .013 | 1.20 | -.040** | 4.16 | - | | - | - | - |
| | PA → EI | - | - | .412** | 39.94 | .412** | 39.94 | - | | - | - | - |
| | NA → EI | - | - | -.171** | 14.74 | -.171** | 14.74 | - | | - | - | - |
| | C → PA | - | - | .383** | 34.42 | .383** | 34.42 | - | | - | - | - |
| | C → NA | - | - | -.190** | 14.89 | -.190** | 14.89 | - | | - | - | - |
| Extraversion model | E → EI | .369** | 37.47 | .352** | 32.66 | .158** | 14.20 | PA: | .161** | 27.40 | .150 | .173 |
| | | | | | | | | NA: | .033** | 10.58 | .027 | .039 |
| | | | | | | | | Total: | .194** | 29.70 | .181 | .206 |
| | Sex → EI | .103** | 8.54 | .069** | 5.85 | .058** | 5.74 | - | | - | - | - |
| | Age→ EI | -.003 | .25 | -.002 | .19 | -.048** | 5.07 | - | | - | - | - |
| | PA → EI | - | - | .432** | 41.78 | .432** | 41.78 | - | | - | - | - |
| | NA → EI | - | - | -.168** | 13.32 | -.168** | 13.32 | - | | - | - | - |
| | E → PA | - | - | .374** | 36.37 | .374** | 36.37 | - | | - | - | - |
| | E → NA | - | - | -.194** | 14.62 | -.194** | 14.62 | - | | - | - | - |
| Agreeableness model | A → EI | .516** | 59.31 | .491** | 48.85 | .356** | 33.79 | PA: | .103** | 19.35 | .093 | .113 |
| | | | | | | | | NA: | .032** | 10.04 | .026 | .039 |
| | | | | | | | | Total: | .135** | 22.09 | .123 | .147 |
| | Sex → EI | .080** | 7.58 | .070** | 6.57 | .044** | 4.72 | - | | - | - | - |
| | Age→ EI | .035* | 3.44 | .031* | 2.95 | -.014 | 1.49 | - | | - | - | - |
| | PA → EI | - | - | .401** | 40.26 | .401** | 40.26 | - | | - | - | - |
| | NA → EI | - | - | -.126** | 11.30 | -.126** | 11.30 | - | | - | - | - |
| | A → PA | - | - | .256** | 21.40 | .256** | 21.40 | - | | - | - | - |
| | A → NA | - | - | -.256** | 16.50 | -.256** | 16.50 | - | | - | - | - |

*(Continued)*

**Table 5.** (Continued)

| | | Direct models | | Mediated models | | | | | | | | |
|---|---|---|---|---|---|---|---|---|---|---|---|---|
| | | Total effect (c) | t-value | Total effect (c) | t-value | Direct effect (c') | t-value | Indirect effect (a.b) | | t-value | LLCI | ULCI |
| Emotional stability model | ES → EI | .521** | 59.83 | .465** | 42.60 | .331** | 26.33 | PA: | .108** | 19.82 | .097 | .119 |
| | | | | | | | | NA: | .027** | 4.09 | .014 | .039 |
| | | | | | | | | Total: | .134** | 15.84 | .118 | .151 |
| | Sex → EI | .092** | 8.09 | .117** | 10.66 | .067** | 6.88 | - | | - | - | - |
| | Age→ EI | -.007 | .64 | -.011 | 1.00 | -.036** | 3.78 | - | | - | - | - |
| | PA → EI | - | - | .427** | 41.34 | .427** | 41.34 | - | | - | - | - |
| | NA → EI | - | - | -.051** | 4.10 | -.051** | 4.10 | - | | - | - | - |
| | ES → PA | - | - | .252** | 21.21 | .252** | 21.21 | - | | - | - | - |
| | ES→ NA | - | - | -.527** | 57.15 | -.527** | 57.15 | - | | - | - | - |

Note: O = Openness to Experience, C = Conscientiousness; E = Extraversion; A = Agreeableness; ES = Emotional Stability; EI = Emotional Intelligence; PA = Positive Affect; NA = Negative Affect; LLCI: Lower Level of Confidence Interval; ULCI: Upper Level of Confidence Interval

**p < .001

*p < .01

.10, thus ensuring that the model explains at least 10% of the variability of the construct [81]. In all five models, the $R^2$ is higher than this level, reaching a level that can be considered moderate. Moderate $R^2$ values are acceptable [71], so it can be affirmed that the five models have adequate predictive power for EI.

The predictive ability of the model is another index used for structural model evaluation. This measure of predictive relevance is based on the Stone-Geisser $Q^2$ statistic [82, 83], which can be measured by blindfolding procedures. According to Henseler and company [71], if this coefficient is greater than 0 for an endogenous latent variable, it means that its explanatory variables provide predictive relevance. In the proposed models, as can be seen in Table 6, this criterion is met.

**Post-hoc analyses of the mediating effects of affect.** The coefficient c, which shows the effect of the direct model of personality traits on emotional intelligence, can be observed in Figs 2–6 and in Table 5. In all of them the total effect is significant and different from zero, indicating a direct relationship [84]. In contrast, the coefficient c' shows the effect of the mediated model, and it can be seen that in all the Figs 2–6 the coefficients of the total effect of personality traits on EI decrease after controlling for the level of positive and negative affect. It is shown, in turn, that the relationship between personality traits and EI remains significant after controlling for the level of affect, suggesting a partial mediation of both positive and negative affect. The bootstrap results show that the indirect effect of personality traits through positive affect, negative affect and both together on EI is statistically significant and different from 0.

**Table 6. Inner model assessment indicators.**

| Model | $R^2$ (IE) | $Q^2$ |
|---|---|---|
| Openness to experience | .368 | .143 |
| Conscientiousness | .363 | .140 |
| Extraversion | .347 | .135 |
| Agreeableness | .431 | .171 |
| Emotional stability | .400 | .155 |

**Table 7. Effect size.**

| Model | Effect | $f^2$ |
|---|---|---|
| **Openness to experience** | Positive affect—IE | .180 |
| | Negative affect—IE | .040 |
| | Openness—IE | .060 |
| **Conscientiousness** | Positive affect—IE | .217 |
| | Negative affect—IE | .042 |
| | Conscientiousness—IE | .056 |
| **Extraversion** | Positive affect—IE | .233 |
| | Negative affect—IE | .039 |
| | Extraversion—IE | .032 |
| **Agreeableness** | Positive affect—IE | .253 |
| | Negative affect—IE | .025 |
| | Agreeableness—IE | .196 |
| **Emotional stability** | Positive affect—IE | .278 |
| | Negative affect—IE | .003 |
| | Emotional stability—IE | .127 |

Note: $f^2$ values of .02, .15 and .35 represent small, medium and large effects respectively

According to the conditions established by Preacher and Hayes [84], affect partially mediates the effect between the Big Five and EI, which partially confirms H4a and H4b hypothesis.

The strength of the effect of positive and negative affect was further assessed for a more detailed analysis of mediation effects. The assessment was done through Cohen's [85] $f^2$ index, which is described as the increase in $R^2$ relative to the proportion of variance of the endogenous latent variable that remains unexplained. Following Cohen's [85] classification of this index, as can be seen in Table 7, in all models, positive affect was found to have medium or medium-high predictive relevance on EI and negative affect, on the other hand, had low or very low relevance. As for each personality trait, openness to experience, conscientiousness and extraversion have low predictive relevance on EI; and medium in the case of agreeableness and emotional stability. In all cases, the strength of the effect of positive affect is greater than the effect of personality traits on EI and the effect of negative affect. This is not the case for negative affect, whose effect is generally of equal significance to that of the personality trait, low. With the exception of agreeableness and emotional stability, whose relevance on EI is greater than that of negative affect.

In conclusion, Big Five, i.e., traits of openness to experience, conscientiousness, extraversion, agreeableness and emotional stability of the researchers are positively related to their emotional intelligence. This relationship is partly explained by, and reinforced through, higher positive affect and lower negative affect, with positive affect having a stronger influence in all cases.

## Discussion

The goals of this empirical study have been to learn the relationships between the five personality traits of the Big Five Factor Model and emotional intelligence in the scientific community, as well as to analyse the mediating role of positive and negative affect in the relationship between the big five personality traits and EI. By doing so, a research gap is filled, since these relationships have not been investigated in the scientific population. Furthermore, the prediction of emotional intelligence in researchers, for example through personal and affective

variables, is crucial because their work includes facing possible frustrations, e.g., not getting the expected results, not gathering representative samples, not being easily published or funded, etc. [2, 37]. Our research has found positive and significant relationships between the big five personality traits, emotional intelligence and positive affect. With negative affects the relationship is negative. Thus, the results show that the five personality traits are positively related to positive affect and negatively related to negative affect in the scientific population. A socially desirable personality scores high in those five traits [86, 87]. The relationships have turned out to form a mediation model, in this case a partial one, so that the socially desirable personality predicts the researcher's EI and that relationship can be explained, in part, by the relationship with affect.

The sample collected was composed of Spanish scientists who had published some research in WoS as it is a comprehensive, reliable, accepted "best practice" database with a large collection of high-impact data [88], which ensured a representative sample of Spanish scientists. Descriptive analyses have led to a description of the personality, affect and EI of researchers in Spain. The group of scientists evaluated tend to manifest high levels of openness to experience, conscientiousness and agreeableness, and slightly lower levels of extraversion and emotional stability. Scientists not only have high scores in openness, as demonstrated by Lounsbury et al. [6], but they also show high scores in conscientiousness, agreeableness, extraversion and emotional stability. Our results back the idea that scientists are well disposed to new experiences and research, and open to new ways of relating concepts. They also tend to be conscientious, emotionally stable, extroverted and agreeable, all of which is useful to strengthen their social relationships in networks. These characteristics, according to previous research, lead to predicting a feeling of satisfaction in their professional work [23] and a better response to situational work demands [2, 16].

The verification of the hypotheses leads to the following debate.

First, as regards Hypothesis 1, the EI of the scientific community maintains positive connections, as a whole, with the five personality traits. People of science who face events with calm and equanimity tend to act in a friendly manner and seek the well-being of others. They are more likely to maintain a high EI, which allows them to better understand their own and others' emotions, so that managing their own emotions helps them be more effective in achieving their goals. All in all, the relations between EI and the five personality traits are generally quite high and are consistent with the conceptualization of EI as a mixture of personality traits [10, 35]. The results of this research complement and enrich the research conducted by Mayer et al. [9], who verified high connections between EI and the personality traits of extraversion and conscientiousness. In our research, in addition to extraversion and conscientiousness, the relationships extend to openness to experience, agreeableness and emotional stability. In the process of achieving objectives in professional performance, scientific people must not only develop cognitive characteristics [8], but also develop socially desirable personality characteristics, manifesting high indices in the big five personality traits [86, 87]. This research adds new information to the results of Araújo et al. [28], who highlighted emotional and motivational processes over personality traits. The big five personality traits will play an important role in observing the degree of professional adaptability. Traits related to curiosity, conscientiousness, openness, friendliness and calmness in the process of getting the job well done can energize mechanisms for the pursuit of professional goals [19]. According to COR theory, the five major personality traits, as a whole, will enable researchers to offer a better response to the demands of research work [16, 19, 89].

Second, regarding hypotheses H2a and H2b, the results from this study show the connections between personality and the positive and negative emotional-affective state of scientific people, as has been confirmed in the previous studies carried out with Russian adult people

[40], Polish people [46], and Australian university students [47]. As expected, researchers' extraversion is associated with positive emotions and greater personal well-being, following the trend of the preceding investigations [6, 42, 45]. Research work involves frequent collaborations and social interactions, so being extroverted can foster these interactions by generating a positive affective state during the work. Not surprisingly, being willing and ready for new experiences and research also translates into more positive emotions and emotional well-being, due to the demands of a researcher's work. Research work requires openness to experience, since an open mind is capable of relating data, contrasting hypotheses and connecting one experience to another, thus promoting positive emotions. On the other hand, emotional stability (or its opposite, neuroticism) has been in the different investigations the strongest correlate of negative affect [42, 45, 48]. In this case, emotional stability of the researchers was found to be strongly and negatively related to negative affect and, consequently, to a negative mood. This finding shows that placing yourself in a state of emotional balance will avoid the subjective discomfort that characterizes negative affect, which is very useful in a job full of situations that can cause frustration. In the same way, the scientific population that tends to manifest a stable emotional state is more likely to be persistent in their work to achieve the goals that lead them to publish their findings. Thus, these results expand previous research that observed the connections of emotional stability with negative mood states and lower job satisfaction [42, 45].

Third, in relation to the third hypothesis in its two variants, the present research confirms the positive relationships between EI and positive affect and the negative relationships between EI and negative affect, following recent previous research results [46, 49, 50]. EI can provide a strong foundation for adaptive resources to help cope with everyday events, in this case, that may occur during research work [2, 3]. In addition, people with high EI are better able to cope with the emotional states that accompany stressful situations in a more harmonious and resilient way. These people are able to analyse events holistically and tend to adopt more functional coping strategies [40, 46]. They are also better able to recognise positive affect. All this, helps them use emotional regulation strategies to deal with events satisfactorily, which are necessary for scientific excellence [1, 33, 40]. Emotional regulation and mood states seem to benefit work behaviour or job performance [2, 35, 44], although mood states come to predict job performance through interpersonal processes (such as helping others or being helped by others) and motivational processes (such as self-efficacy in the task) [88]. These results are of special interest in the present sample of researchers because their work is exposed to many frustrations, so emotional control can play a key role [51].

Fourth, the results have confirmed that positive affect and negative affect show significant indirect effects on the relationship between each of the big five personality traits and EI, with the latter variables also retaining their direct effects. It can be said that maintaining a high level of positive affect will increase the relationship between personality traits and EI, whereas negative affect will act as a brake in the relationship between personality traits and EI. In fact, the present study shows that mediating models explain about 35% of the variability of EI, which is quite a high percentage and demonstrates the importance of these results. As can be seen in the resulting models, the paths established in the direct relationships between each of the personality traits and emotional intelligence are significant (Figs 2–6). However, when taking into account the positive and negative affect as mediators, the significance of the effect of personality traits on emotional intelligence decreases, and the $R^2$ increases, reaching up to 43.1% of the variance in the case of agreeableness. Hence, each one of the five personality traits stimulates EI, especially when they are linked to positive affective states [43]. People prone to maintaining positive affective states, coupled with a more socially desirable personality [86, 87], are more likely to be able to manage their own and others' emotional states, optimising job performance

[35]. Positive emotions encourage regulatory resources in the course of emotional experiences. In this way, they help to counteract the adverse physiological effects derived from negative affect [16]. This is more likely to occur if the researcher is: extroverted and friendly in the continuous course of collaborations and social relationships involved in her work; responsible and organized in her daily professional and personal duties; open to new situations and experiences that allow her to keep on generating novel research; and emotionally stable in a job in which she is often exposed to frustration. Thus, both the aforementioned personality traits and affective and emotional characteristics improve flexible thinking and attention, which are necessary for job performance [39, 42, 44].

Fifth, it should also be noted that the contribution of positive affect in predicting EI is significantly greater than that of negative affect. That is, maintaining positive affect contributes to a greater extent to achieving better management of their emotions than avoiding negative affect. A positive mood in researchers will encourage a more positive interpretation in evaluating day-to-day events [11]. Furthermore, it will help to develop resources to regulate negative emotions, improving behavioural flexibility and boosting well-being [39, 42].

Thus, researchers who are capable of handling emotional states, fostering positive ones as opposed to negative ones, and who also show socially desirable personality traits, will be better able to deal with emotions harmoniously and tackle day-to-day and work occurrences positively. All this will contribute to experiencing greater subjective well-being [10, 46, 47], which could increase performance at research work [2, 35].

To sum up, the results obtained in this study show that there are relationships between personality and EI, and they highlight the partial mediating role of positive and negative affect on research and, by extension, on researchers who publish in the journals included in the WoS. These results contribute with valuable information about the central role of the big five personality traits in in job performance and shed light on emotional factors in the scientific community. The significance of positive emotions in building resources to regulate negative emotional experiences is affirmed, helping mitigate a lowering of behavioural repertoire more typical of negative emotions [11, 51]. Consequently, is crucial to take such strategies into account to foster scientific excellence and the work of the scientific population [8, 33]. Hence, the role of emotional factors and personality traits in scientific excellence is confirmed. It should not be forgotten that knowledge generation in today's society is a motor for economic and social development [8, 18]. These findings provide information on the importance of fostering positive affect in organizations as well as EI in researchers, given the importance of emotions in their work. The influence, although small, of avoiding negative affect or, in a more practical way, teaching how to cope with it, should also be taken into account [51]. These results may also raise awareness among researchers of the need to prioritise wellbeing and mental health in order to achieve scientific excellence.

## Limitations

This study has had some limitations. One limitation is related to the cross-sectional nature of the research, which does not enable causal relationships to be established. A longitudinal study should be carried out in the future to extend the results. Secondly, the questionnaires as a whole are part of a larger project and they contain about 160 items, so it is possible that there was a bias due to the effect of fatigue or withdrawal. In the latter case, all of the questionnaires that were not completely filled in were eliminated. Another limitation may arise from the evaluation questionnaires themselves aimed at analysing the variables of this empirical work. For this reason, questionnaires and scales have been used that have demonstrated their reliability and validity in previous studies. Moreover, the study was based on corresponding authors

from publications in the WoS database, thereby establishing their scientific excellence. WoS has been considered to be a database that provides greater confidence on recognising the scientific professionalism of the publications in it. Nevertheless, this is questionable and may require a cut-off with stricter criteria. On the other hand, as regards possible future studies, it would be interesting to study if socio-economic status shapes emotional intelligence. Moreover, personality traits and emotions interact in complex ways with technological innovation and pro-environmental behaviour, shaping individuals' attitudes, motivations and actions towards technology and the environment [90], and future studies that deepen our understanding of these relationships may provide insights into how to promote sustainable behaviours and encourage the adoption of environmentally friendly technologies.

## Conclusion

In light of the results, it can be concluded that, while personality traits have a fairly large connection with EI, this connection is enhanced by affect, primarily positive affect. The ability to manage emotional states will be influenced, to a large extent, by the personality traits of the researchers. However, the action of positive affect will enhance the relationship between each of the big five personality traits and EI, whereas negative affect will diminish the relationship between personality traits and EI. High EI scores indicate better ability to manage emotions and to guide thoughts and actions effectively [3, 31].

To some extent, the results support the COR theory of personality response as a personal resource that facilitates more adaptive performance to demands [16]. In turn, positive affect will act as an enhancer and help regulate emotional experiences to improve the ability to manage the emotional system. Furthermore, according to Fredrickson's contribution [11], positive emotions help to regulate negative emotions. As known, EI is important in coping with daily life events that have a response on job performance [19, 89].

These findings have the potential to hold significant practical implications. They can significantly influence the contributions of the scientific population to the development of science, helping to cultivate emotionally intelligent scientific practices, thereby improving the overall impact of the scientific community and contributing to the advancement of society.

From a practical perspective, these confirmations contribute with information for designing programmes aimed at fostering scientific excellence. The extensive sample size of this study enables the findings to be applicable various settings, including universities, research organisations, healthcare facilities, businesses and other organisations across a wide range of scientific fields (technological, medical, life, natural and social sciences). The findings are highly likely to be relevant to researchers who were not included in our sample, as they experience comparable pressures. This includes researchers in countries, languages or science fields that are underrepresented in the WoS, as well as those working in organisations where publishing holds less significance, such as businesses or non-governmental organizations.

These results offer insights into the importance of understanding and managing emotions, as well as the beneficial role of positive affects. Implementing strategies to promote positive affects and mitigate negative ones is crucial. Researchers often encounter frustrating situations such as the rejection of their work by the scientific community despite their significant efforts. This inability to publish their findings can be emotionally burdensome. Therefore, in addition to considering the influence of the big five personality traits, it is recommended to take into account EI and positive affects when addressing the emotional challenges associated with such setbacks.

Thus, the importance of emotions in the work of researchers should be taken into account, training EI, but also encouraging positive affect as much as possible and training in

appropriate coping with negative affect. These findings could also raise awareness among researchers of the importance of prioritising wellbeing and mental health for scientific excellence.

On the other hand, this information can help in the process of searching for the most suitable profile when selecting research staff and to design training programs for current employees. In addition, it should be noted that the effectiveness of interventions has been shown to vary according to individual differences, with personality playing a major role [91]. Thus, some actions can be very advantageous for research organizations: encouraging positive affect with actions that motivate researchers; minimizing, as much as possible, negative affect; encouraging emotional regulation skills, and having previously assessed personality traits to advocate for researchers with socially desirable personality characteristics. Furthermore, although personality is relatively stable, employees can be trained in or encouraged to engage in behaviours typically associated with that socially desirable personality, as proposed by Rubino et al. [16].

## Acknowledgments

Our special thanks go to the volunteers who pilot-tested the survey: David Barberá, Àngels Bernabeu, Gérard Carat, María Ángeles Chavarría, Ester Linde, Óscar Llopis, Francisco Rivas and Soberana Sáez. Thanks are due also to the survey respondents and, particularly, those who spontaneously provided supporting statements.

## Author Contributions

**Conceptualization:** Laura Hernando-Jorge, Ana M. Tur-Porcar.

**Formal analysis:** Laura Hernando-Jorge, Anabel Fernández-Mesa.

**Funding acquisition:** Joaquín M. Azagra-Caro.

**Investigation:** Anabel Fernández-Mesa.

**Methodology:** Laura Hernando-Jorge, Anabel Fernández-Mesa, Ana M. Tur-Porcar.

**Resources:** Laura Hernando-Jorge, Joaquín M. Azagra-Caro, Ana M. Tur-Porcar.

**Supervision:** Anabel Fernández-Mesa, Joaquín M. Azagra-Caro, Ana M. Tur-Porcar.

**Writing – original draft:** Laura Hernando-Jorge.

**Writing – review & editing:** Laura Hernando-Jorge, Anabel Fernández-Mesa, Joaquín M. Azagra-Caro, Ana M. Tur-Porcar.

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
