## [Decision Letter · Decision Letter 0]

25 Apr 2023

PONE-D-23-02627Personality and emotional intelligence of researchers: the importance of affectsPLOS ONE

Dear Dr. Hernando,

Thank you for submitting your manuscript to PLOS ONE. After careful consideration, we feel that it has merit but does not fully meet PLOS ONE’s publication criteria as it currently stands. Therefore, we invite you to submit a revised version of the manuscript that addresses the points raised during the review process.

We look forward to receiving your revised manuscript.

Kind regards,

Sohaib Mustafa

Academic Editor

PLOS ONE

Journal Requirements:

2. Please provide additional details regarding ethical approval in the body of your manuscript. In the Methods section, please ensure that you have specified the name of the IRB/ethics committee that approved your study.

4. We noted in your submission details that a portion of your manuscript may have been presented or published elsewhere. "An earlier version of this paper was presented at the V National Congress of Psychology and International Symposium on Public Health Psychology (CNP2021), Online, July 9-11, 2021. A preprint can be found in one of our affiliation's working paper repository: https://www2.ingenio.upv.es/es/working-papers/personality-and-affects-researchers-emotional-intelligence . The submission introduces a major change of methodology, from regression analysis to structural equation modelling, with consequent adaptations throughout the text. The change is so substantial that the submission includes an additional author, Anabel Fernández-Mesa, expert in structural equation modelling." Please clarify whether this [conference proceeding or publication] was peer-reviewed and formally published. If this work was previously peer-reviewed and published, in the cover letter please provide the reason that this work does not constitute dual publication and should be included in the current manuscript.

Additional Editor Comments:

Your work is of great potential, but reviewers found some loopholes which can be improved to highlight your findings further.

Reviewer noticed that your work lacks in support of arguments. I suggest you cite and compare your work findings to recently published work on big five personality traits to reflect better and explain your findings. You can consider articles such as,

https://www.tandfonline.com/doi/abs/10.1080/10447318.2023.2201551

https://doi.org/10.3389/fpsyg.2022.956281

Reviewers' comments:

Reviewer's Responses to Questions

**Comments to the Author**

1. Is the manuscript technically sound, and do the data support the conclusions?

Reviewer #1: Yes

Reviewer #2: Partly

2. Has the statistical analysis been performed appropriately and rigorously? 

Reviewer #1: Yes

Reviewer #2: I Don't Know

3. Have the authors made all data underlying the findings in their manuscript fully available?

Reviewer #1: No

Reviewer #2: Yes

4. Is the manuscript presented in an intelligible fashion and written in standard English?

Reviewer #1: Yes

Reviewer #2: Yes

5. Review Comments to the Author

Reviewer #1: The article is indeed very interesting and thouroughly performed, but in my opinion the authors should focus better on outlining the importance of their study instead of describing the review of the literature since this is a research article. Therefore, please remove the unnecessary details from the first section and mention only an introduction regarding the current study.

Reviewer #2: Dear Author,

Thank you for the opportunity to review the manuscript „Personality and emotional intelligence of researchers: the importance of affects”.

This study analysis the relationships between personality traits, emotional intelligence (EI) and positive and negative affect among the scientific population.

I make the following recommendations to improve the manuscript:

1. ABSTRACT: The abstract should be concise and informative.It should briefly describe the purpose of the work, techniques and methods used, major findings with important data and conclusions. Please edit the aim of the research. Briefly mention the main aim of the work and then write the specific aim.

2. Cite references in brackets (for example, “[1]” or “[2-5]” or “[3,7,9]”).

3. INTRODUCTION: Please improve introduction.

a. Introduction should be shorter and has the most important informations. Please explain why this research is so important and why you want to fill the gaps about the personality and emotional intelligence of researchers.

b. Why you made so many subsections?

c. Please clearly define the aim of the work and the the hypothesis.

4. MATERIAL AND METHOD: Please improve this section,

a. Participants:

b. Why did you contact only with corresponding authors for publications appearing in the Web of Science (WoS) from 2013 to 2016? Why did you choose these years?

c. Why was you not ask socioeconomic status?

d. Please rewrite the characteristic of the respondents.

e. Please add: exlusion and inclusion criteria

f. It seems necessary to clarify how the representative sample was calculated and to make a flowchart of the enrollment of respondents and study procedures. You should explain the sample size calculation in order to know if this sample is representative.

g. Statistical Analysis: please improve this section. Did you use only Pearson correlation? The main issue of the paper is that, the description of the statistical analysis plan is unclear. While there are many results that are presented, only one paragraph was devoted to the statistical analysis. The authors need to pay more attention into the details of analysis plan in order to make the paper scientifically rigorous. Please add all information about analysis in this section, not in the results.

h. The design of the study is not explained, please describe it.

i. I think you should write a section related to Ethical Considerations. You should clarify the ethics of your research. Please add sentence: „The study was conducted according to the guidelines of the Declaration of Helsinki and approved by the Ethics Committee of the …”.

j. RESULTS:

i. each of the results should be mentioned in more depth and more readable for the reader.

k. DISCUSSION: Please improve this section.

i. The discussion of the results should be done in a more in-depth way.

ii. The discussion should discuss the implications of the findings in general and be supported by relevant and related articles.

iii. I miss an introductory paragraph in which the aim of the research and the main results found are remembered. In general, the wording of the discussion could be better. I suggest preparing some kind of summary with gaps indication and future perspective

l. CONCLUSION: The conclusion must include a summary of the key ideas and the most important issues raised throughout the research paper. Please add this subsection!

m. LIMITATION: please add limitations like subsection. Please discuss widely the advantages and disadvantages of this research, identify knowledge gaps in the existing literature, and try to indicate the needs for future research.

6. PLOS authors have the option to publish the peer review history of their article (what does this mean?). If published, this will include your full peer review and any attached files.

Reviewer #1: No

Reviewer #2: No

---

## [Author Response · Author response to Decision Letter 0]

23 Jun 2023

We have included a file with the response to each point raised by both the editor and the reviewers. We have also uploaded the images to the digital diagnostic tool Preflight Analysis and Conversion Engine (PACE) and re-uploaded them.

---

## [Decision Letter · Decision Letter 1]

22 Jan 2024

PONE-D-23-02627R1Personality and emotional intelligence of researchers: the importance of affectsPLOS ONE

Dear Dr. Hernando,

Thank you for submitting your manuscript to PLOS ONE. After careful consideration, we feel that it has merit but does not fully meet PLOS ONE’s publication criteria as it currently stands. Therefore, we invite you to submit a revised version of the manuscript that addresses the points raised during the review process.

We look forward to receiving your revised manuscript.

Kind regards,

Sohaib Mustafa, Ph.D.

Academic Editor

PLOS ONE

Reviewers' comments:

Reviewer's Responses to Questions

**Comments to the Author**

1. If the authors have adequately addressed your comments raised in a previous round of review and you feel that this manuscript is now acceptable for publication, you may indicate that here to bypass the “Comments to the Author” section, enter your conflict of interest statement in the “Confidential to Editor” section, and submit your "Accept" recommendation.

Reviewer #3: (No Response)

Reviewer #4: (No Response)

2. Is the manuscript technically sound, and do the data support the conclusions?

Reviewer #3: Yes

Reviewer #4: Yes

3. Has the statistical analysis been performed appropriately and rigorously? 

Reviewer #3: Yes

Reviewer #4: Yes

4. Have the authors made all data underlying the findings in their manuscript fully available?

Reviewer #3: Yes

Reviewer #4: No

5. Is the manuscript presented in an intelligible fashion and written in standard English?

Reviewer #3: No

Reviewer #4: Yes

6. Review Comments to the Author

Reviewer #3: This manuscript delves into the crucial role that the Big Five personality traits play in the emotional intelligence of researchers. The subject matter of this study is intriguing, and a substantial dataset has been employed to validate the article’s hypothesis. However, the reviewers have identified several areas for improvement:

1. The introduction of the manuscript could benefit from a more thorough explanation. This should underscore the research question posed by the article, the objectives of the research, and the significance of the study.

2. The article would be strengthened by a more exhaustive review of the relevant literature, which would help to substantiate the hypotheses presented.

3. Given that Plos One is a multidisciplinary journal with a diverse readership, it would be beneficial for the author to dedicate a few paragraphs to introducing the concept of personality traits. Additionally, recent comprehensive discussions on technological innovation and pro-environmental behavior should be included. This would enable readers to better comprehend the importance of this study.

4. The manuscript’s academic writing style requires enhancement. The use of subjective pronouns such as “we” and “I”, as well as subjective expressions like “may be”, should be avoided to maintain a professional tone throughout the article.

5. In the “design of the study” section, the author should provide a more detailed account of the data collection process. This could include information about the platform used to distribute the questionnaire. It would also be helpful to report some basic demographic characteristics of the respondents, such as their tenure in the research industry and their roles within it (e.g., graduate student, junior scholar, senior scholar, manager, etc.).

Reviewer #4: It’s an interesting study however authors are needed to incorporate certain changes to make this study publishable.

1. The introduction section of the study needs improvement. The research background (common ground) and gap (complexity) are unclearly covered. Please allow me to offer an alternative and in my opinion, more intuitive and attractive way of structuring an introduction. I suggest that the authors follow the suggestions of Lange and Pfarrer, (2017) when structuring the introduction. Such structure allows completing the introduction with five major paragraphs with central story line.

Lange,D., & Pfarrer, M.D. (2017). Editors' comments: Sense and structure- The core building blocks of an AMR article. Academy of Management Review, 42(3), 407-416

2. I would also suggest to cite additional relevant literature which demonstrate the affective health of knowledge workers that is crucial for the betterment of researchers' affective and practical well-being cum enhanced performance.

1. Shafait, Z., & Huang, J. (2023). Exploring the Nexus of Emotional Intelligence and University Performance: An Investigation Through Perceived Organizational Support and Innovative Work Behavior. Psychology Research and Behavior Management, 4295-4313.

3. Literature/ use of theory. As a rule of thumb, make the lit review about the same length as the discussion. But you cannot make this more than 1800 words long, to stay within your word limit. In terms of the content, you need to go beyond describing a series of relevant references and tell us how your interpretation of the literature shows the gaps that exist, and how the proposed approach to the literature brings about novel opportunities to reinterpret the literature that will allow advancement in our understanding in the field. The literature review is too descriptive, and I cannot see the wood for the trees.

Please work on your theoretical underpinnings. How theory can be fixed with these variables and can contribute. Incorporating the “WHY” factor is important.

4. Methodology. You need to better justify your choice of methodology, and whether this is either innovative or the standard approach, for your theoretical framework (bearing in mind that the theoretical framework is not clearly defined).

5. Results and discussion. While the results are rigorous, their interpretation is quite descriptive and lacks originality. The format of your writing means that you are playing safe here, for example, you mechanically show how your findings corroborate aspects of the literature that were already well-known to the readers. Instead, you ought to be more ambitious here, and show what’s really new about your findings, but also how does your work expand the current understanding of what the literature has already reported, how you are saying something that makes a substantial change in our collective understanding that all authors studying this field should be aware of and cite in the future. Your contribution needs to be much more explicit; your discussion needs to genuinely expand the boundaries of knowledge.

6. Discussion. This needs to be a coherent and cohesive set of arguments that take us beyond this study in particular and help us see the relevance of what you have found to the wider world. You need to emphasize the significance of your work: How does your finding help us transform researchers towards being more mental-health friendly?

7. Conclusions. This section needs more than a summary of the previous sections and adds value. You need to emphasize and explicitly spell out the contribution to the theory that your analysis brings about.

Discussion should be done considering the tested hypotheses. Please read and cite the latest papers as there are a lot of recent papers on this topic on the Web of Science.

Revisit your theoretical implication. Didn’t find any theory and you have concluded your theoretical implications.

7. PLOS authors have the option to publish the peer review history of their article (what does this mean?). If published, this will include your full peer review and any attached files.

Reviewer #3: No

Reviewer #4: No

---

## [Author Response · Author response to Decision Letter 1]

6 Mar 2024

March 6th, 2024

Dear Professor Sohaib Mustafa, 

Thank you for your invitation to revise and resubmit our paper in this second round of revision and for giving us detailed instructions about how to perform the revision.

We appreciate all the comments from reviewers 3 and 4, they have provided valuable feedback which has helped us to improve our paper. 

We hope to have fully addressed all your concerns and the concerns by the two anonymous reviewers from the first round and from reviewers 3 and 4 from the second round in this new version of the manuscript. 

Best regards,

The authors

 

 TO THE ATTENTION OF REVIEWER 3

General comment

This manuscript delves into the crucial role that the Big Five personality traits play in the emotional intelligence of researchers. The subject matter of this study is intriguing, and a substantial dataset has been employed to validate the article’s hypothesis. However, the reviewers have identified several areas for improvement.

Response. Thank you very much for reading our paper and giving us detailed advice about how to improve it. We really appreciate this opportunity. We hope to have properly addressed your concerns. Concretely, in this new version of the manuscript we have updated the introduction section, the literature review and hypotheses, included some paragraphs on the concept of personality traits and pro-environmental behaviour and technological innovation (these last two topic as a promising future research). We have changed the use of pronouns we and I, as well as subjective expressions like “may be”. And finally, we have also included more detail of the design of the study. 

Comment #1

The introduction of the manuscript could benefit from a more thorough explanation. This should underscore the research question posed by the article, the objectives of the research, and the significance of the study.

Response. Thank you for the recommendation. We have rewritten and reorganised the introduction accordingly, trying to make it more comprehensive, making the problem posed and the proposed objectives more evident.

Comment #2

The article would be strengthened by a more exhaustive review of the relevant literature, which would help to substantiate the hypotheses presented.

Response. We appreciate your suggestion. We have reviewed the literature again and tried to restructure and improve the literature review section as proposed. 

Comment #3

Given that Plos One is a multidisciplinary journal with a diverse readership, it would be beneficial for the author to dedicate a few paragraphs to introducing the concept of personality traits. Additionally, recent comprehensive discussions on technological innovation and pro-environmental behavior should be included. This would enable readers to better comprehend the importance of this study.

Response. Thank you very much for bringing this issue to our attention. The information on personality traits has been expanded in the literature review section, and their relationship to technological innovation and pro-environmental behaviour has also been included as a possibility for future study in the relevant section.

Comment #4

The manuscript’s academic writing style requires enhancement The use of subjective pronouns such as “we” and “I”, as well as subjective expressions like “may be”, should be avoided to maintain a professional tone throughout the article.

Response. We are very grateful for your comment. We have examined the writing style throughout the article and tried to improve the tone as much as possible by applying your suggestions. We have also hired an English reviewer to edit the whole manuscript. 

Comment #5

In the “design of the study” section, the author should provide a more detailed account of the data collection process. This could include information about the platform used to distribute the questionnaire. It would also be helpful to report some basic demographic characteristics of the respondents, such as their tenure in the research industry and their roles within it (e.g., graduate student, junior scholar, senior scholar, manager, etc.).

Response. The platform used was Qualtrics. We prefer not to disclose it in the paper to avoid promotion of a private business firm who is already charging a license fee to our organisation for the use of its proprietary software.

We included demographic characteristics regarding organisation type and tenure. We included a self-citation to a report with the full demographics –anonymised for the revision process.

Thank you for your valuable and beneficial feedback, which we believe contributed to enhancing the paper. We are willing and eager to implement additional revisions or address any further concerns you may have.

TO THE ATTENTION OF REVIEWER 4

General comment

It’s an interesting study however authors are needed to incorporate certain changes to make this study publishable.

Response. Thank you very much for your time and effort. Your comments were very clear and detailed, facilitating us the inclusion of your suggestions. We really appreciate your constructive tone and advise. 

Comment #1

The introduction section of the study needs improvement. The research background (common ground) and gap (complexity) are unclearly covered. Please allow me to offer an alternative and in my opinion, more intuitive and attractive way of structuring an introduction. I suggest that the authors follow the suggestions of Lange and Pfarrer, (2017) when structuring the introduction. Such structure allows completing the introduction with five major paragraphs with central story line.

Lange,D., & Pfarrer, M.D. (2017). Editors' comments: Sense and structure- The core building blocks of an AMR article. Academy of Management Review, 42(3), 407-416

Response. Thank you for your comments and suggested literature. We have included your recommendations in the introduction by reorganising and completing the details proposed in the suggested article.

Comment #2

I would also suggest to cite additional relevant literature which demonstrate the affective health of knowledge workers that is crucial for the betterment of researchers' affective and practical well-being cum enhanced performance.

Shafait, Z., & Huang, J. (2023). Exploring the Nexus of Emotional Intelligence and University Performance: An Investigation Through Perceived Organizational Support and Innovative Work Behavior. Psychology Research and Behavior Management, 4295-4313.

Response. We are very grateful for this comment and, again, for the suggested bibliography. We have proceeded accordingly by checking for new, more recent bibliography such as the one you suggest, making the research more up to date.

Comment #3

Literature/ use of theory. As a rule of thumb, make the lit review about the same length as the discussion. But you cannot make this more than 1800 words long, to stay within your word limit. In terms of the content, you need to go beyond describing a series of relevant references and tell us how your interpretation of the literature shows the gaps that exist, and how the proposed approach to the literature brings about novel opportunities to reinterpret the literature that will allow advancement in our understanding in the field. The literature review is too descriptive, and I cannot see the wood for the trees.

Please work on your theoretical underpinnings. How theory can be fixed with these variables and can contribute. Incorporating the “WHY” factor is important.

Response. Thank you for your feedback. We have re-written this section accordingly, so that the introduction and literature is now very similar in length to the discussion, we have tried to go further with the explanations and worked on the theoretical foundations as much as possible.

Comment #4

Methodology. You need to better justify your choice of methodology, and whether this is either innovative or the standard approach, for your theoretical framework (bearing in mind that the theoretical framework is not clearly defined).

Response. Thank you for noticing this. We have included a justification of the standard quantitative approach followed and the choice of survey and structural equation modelling methods.

Comment #5

Results and discussion. While the results are rigorous, their interpretation is quite descriptive and lacks originality. The format of your writing means that you are playing safe here, for example, you mechanically show how your findings corroborate aspects of the literature that were already well-known to the readers. Instead, you ought to be more ambitious here, and show what’s really new about your findings, but also how does your work expand the current understanding of what the literature has already reported, how you are saying something that makes a substantial change in our collective understanding that all authors studying this field should be aware of and cite in the future. Your contribution needs to be much more explicit; your discussion needs to genuinely expand the boundaries of knowledge.

Response. Thank you very much for your comment. To improve this section, it has been rewritten taking into account your suggestions, trying to show more ambitiously the implications of the results and what they really imply.

Comment #6

Discussion. This needs to be a coherent and cohesive set of arguments that take us beyond this study in particular and help us see the relevance of what you have found to the wider world. You need to emphasize the significance of your work: How does your finding help us transform researchers towards being more mental-health friendly?

Response. We have included your suggestions in the section to improve the clarity and relevance of the results in society. We have also emphasised how this helps researchers to be more mental-health friendly.

Comment #7

Conclusions. This section needs more than a summary of the previous sections and adds value. You need to emphasize and explicitly spell out the contribution to the theory that your analysis brings about.

Discussion should be done considering the tested hypotheses. Please read and cite the latest papers as there are a lot of recent papers on this topic on the Web of Science.

Revisit your theoretical implication. Didn’t find any theory and you have concluded your theoretical implications.

Response. Thank you for all your suggestions. We have tried to add value in the conclusion by adding ideas that make the contributions of our results more explicit. We have also added new, more recent citations and revised the theoretical contribution included.

Thank you for your constructive and helpful comments, which, in our view, helped improve the paper. We would be more than happy to make further changes/respond to any additional concerns that you may have.

---

## [Decision Letter · Decision Letter 2]

24 Apr 2024

PONE-D-23-02627R2Personality and emotional intelligence of researchers: the importance of affectsPLOS ONE

Dear Dr. Hernando,

Thank you for submitting your manuscript to PLOS ONE. After careful consideration, we feel that it has merit but does not fully meet PLOS ONE’s publication criteria as it currently stands. Therefore, we invite you to submit a revised version of the manuscript that addresses the points raised during the review process.Please submit your revised manuscript by Jun 08 2024 11:59PM. If you will need more time than this to complete your revisions, please reply to this message or contact the journal office at plosone@plos.org. Please include the following items when submitting your revised manuscript:A rebuttal letter that responds to each point raised by the academic editor and reviewer(s). You should upload this letter as a separate file labeled 'Response to Reviewers'.A marked-up copy of your manuscript that highlights changes made to the original version. You should upload this as a separate file labeled 'Revised Manuscript with Track Changes'.An unmarked version of your revised paper without tracked changes. You should upload this as a separate file labeled 'Manuscript'.If applicable, we recommend that you deposit your laboratory protocols in protocols.io to enhance the reproducibility of your results. Protocols.io assigns your protocol its own identifier (DOI) so that it can be cited independently in the future. For instructions see: https://journals.plos.org/plosone/s/submission-guidelines#loc-laboratory-protocols. Additionally, PLOS ONE offers an option for publishing peer-reviewed Lab Protocol articles, which describe protocols hosted on protocols.io. Read more information on sharing protocols at https://plos.org/protocols?utm_medium=editorial-email&utm_source=authorletters&utm_campaign=protocols.

We look forward to receiving your revised manuscript.

Kind regards,

Sohaib Mustafa, Ph.D.

Academic Editor

PLOS ONE

Journal Requirements:

Reviewers' comments:

Reviewer's Responses to Questions

**Comments to the Author**

1. If the authors have adequately addressed your comments raised in a previous round of review and you feel that this manuscript is now acceptable for publication, you may indicate that here to bypass the “Comments to the Author” section, enter your conflict of interest statement in the “Confidential to Editor” section, and submit your "Accept" recommendation.

Reviewer #3: (No Response)

Reviewer #4: (No Response)

2. Is the manuscript technically sound, and do the data support the conclusions?

Reviewer #3: Yes

Reviewer #4: (No Response)

3. Has the statistical analysis been performed appropriately and rigorously? 

Reviewer #3: Yes

Reviewer #4: (No Response)

4. Have the authors made all data underlying the findings in their manuscript fully available?

Reviewer #3: Yes

Reviewer #4: (No Response)

5. Is the manuscript presented in an intelligible fashion and written in standard English?

Reviewer #3: Yes

Reviewer #4: (No Response)

6. Review Comments to the Author

Reviewer #3: The reviewers believe that the manuscript has been revised and improved significantly by the authors. Before recommending acceptance of the manuscript, the reviewer also has a small suggestion for the authors to consider: the manuscript will further highlight the need for the research question, especially whether the group of academic workers is representative and whether the findings of this study have chance. Be extended to more groups and have possible contributions to application-based applications.

Reviewer #4: Authors have responded to the raised concerns comprehensively; therefore, I suggest acceptance of this manuscript in current form.

Best wishes for authors

7. PLOS authors have the option to publish the peer review history of their article (what does this mean?). If published, this will include your full peer review and any attached files.

Reviewer #3: No

Reviewer #4: No

---

## [Author Response · Author response to Decision Letter 2]

17 May 2024

TO THE ATTENTION OF REVIEWER 3

General comment

The reviewers believe that the manuscript has been revised and improved significantly by the authors. 

Response. We are glad to know that you have notice the improvement. Thank you to help us to bring the paper to a higher level of quality.

Comment #1

Before recommending acceptance of the manuscript, the reviewer also has a small suggestion for the authors to consider: the manuscript will further highlight the need for the research question, especially whether the group of academic workers is representative and whether the findings of this study have chance. Be extended to more groups and have possible contributions to application-based applications.

Response. You are totally right. We have better clarified the gap and aim of the study in the introduction. 

P. 3 (introduction): 

“At the core of managing these emotional complexities lies emotional intelligence (EI), essential for regulating one´s emotions and navigating challenging situations (1). In particular, researchers must constantly manage intense feelings of excitement, frustration, and self-doubt as they pursue discoveries, secure funding, and face the peer review process. Possessing high EI can help researchers maintain focus, foster productive collaborations, and persevere through the setbacks inherent in scientific research [5]. 

Despite the importance of EI in research settings, our understanding of researchers' emotional intelligence remains limited [3], failing to reflect the societal significance of these knowledge generators. Researchers often exhibit unique personality traits, such as introversion and neuroticism [6, 7], which may pose challenges in developing and utilizing emotional intelligence. Addressing this gap in the literature is essential, as enhancing researchers' EI could lead to improved well-being, increased research productivity, and more impactful scientific contributions.”

Additionally, we have clarified the wide representation of researchers from every discipline in the methods (p. 11):

The distribution by science field is as follows: technological sciences (14%), medical sciences (24%), life sciences (16%), natural sciences (24.48%) and social sciences (21%). This broad representation of researchers underscores the diversity of participants in the study.

The sample includes researchers that have published WoS papers (working is universities, research organisations, health establishments, business firms, public administrations and non-governmental organisations). But the results could apply to researchers that do not publish WoS papers. So, in the conclusions we have highlighted the representativeness of our sample and clarified that the results could also be extended to other groups. 

P. 31 and 32: 

“From a practical perspective, these confirmations contribute with information for designing programmes aimed at fostering scientific excellence. The extensive sample size of this study enables the findings to be applicable various settings, including universities, research organisations, healthcare facilities, businesses and other organisations across a wide range of scientific fields (technological, medical, life, natural and social sciences). The findings are highly likely to be relevant to researchers who were not included in our sample, as they experience comparable pressures. This includes researchers in countries, languages or science fields that are underrepresented in the WoS, as well as those working in organisations where publishing holds less significance, such as businesses or non-governmental organizations.

These results offer insights into the importance of understanding and managing emotions, as well as the beneficial role of positive affects. Implementing strategies to promote positive affects and mitigate negative ones is crucial. Researchers often encounter frustrating situations such as the rejection of their work by the scientific community despite their significant efforts. This inability to publish their findings can be emotionally burdensome. Therefore, in addition to considering the influence of the big five personality traits, it is recommended to take into account EI and positive affects when addressing the emotional challenges associated with such setbacks.”

Thank you for your insightful and helpful input, which we believe has helped improve the paper. We hope you will find the article ready for publication, but if you consider that it still needs any adjustment, we are eager to implement any additional concern you may have.

TO THE ATTENTION OF REVIEWER 4

General comment

Authors have responded to the raised concerns comprehensively; therefore, I suggest acceptance of this manuscript in current form. 

Response. Thank you sincerely for granting us the chance to enhance the paper and aiding us in crafting a revised edition of the manuscript. We are deeply grateful for the opportunity you've given us to publish the article.

---

## [Decision Letter · Decision Letter 3]

21 May 2024

Personality and emotional intelligence of researchers: the importance of affects

PONE-D-23-02627R3

Dear Dr. Hernando,

We’re pleased to inform you that your manuscript has been judged scientifically suitable for publication and will be formally accepted for publication once it meets all outstanding technical requirements.

Kind regards,

Sohaib Mustafa, Ph.D.

Academic Editor

PLOS ONE

Additional Editor Comments (optional):

Reviewers' comments:

Reviewer's Responses to Questions

**Comments to the Author**

1. If the authors have adequately addressed your comments raised in a previous round of review and you feel that this manuscript is now acceptable for publication, you may indicate that here to bypass the “Comments to the Author” section, enter your conflict of interest statement in the “Confidential to Editor” section, and submit your "Accept" recommendation.

Reviewer #3: (No Response)

2. Is the manuscript technically sound, and do the data support the conclusions?

Reviewer #3: (No Response)

3. Has the statistical analysis been performed appropriately and rigorously? 

Reviewer #3: (No Response)

4. Have the authors made all data underlying the findings in their manuscript fully available?

Reviewer #3: (No Response)

5. Is the manuscript presented in an intelligible fashion and written in standard English?

Reviewer #3: (No Response)

6. Review Comments to the Author

Reviewer #3: (No Response)

7. PLOS authors have the option to publish the peer review history of their article (what does this mean?). If published, this will include your full peer review and any attached files.

Reviewer #3: No

---

## [Editor Report · Acceptance letter]

3 Jul 2024

PONE-D-23-02627R3 

PLOS ONE

Dear Dr. Hernando-Jorge, 

I'm pleased to inform you that your manuscript has been deemed suitable for publication in PLOS ONE. Congratulations! Your manuscript is now being handed over to our production team.

Kind regards, 

on behalf of

Dr. Sohaib Mustafa 

Academic Editor

PLOS ONE